# Fabrication Methods for Microfluidic Devices: An Overview

**DOI:** 10.3390/mi12030319

**Published:** 2021-03-18

**Authors:** Simon M. Scott, Zulfiqur Ali

**Affiliations:** Healthcare Innovation Centre, School of Health and Life Sciences, Teesside University, Middlesbrough, Tees Valley TS1 3BX, UK

**Keywords:** microfluidics, micro- and nanofabrication, micromachining, hot embossing, injection moulding, laminate, laser ablation, 3D printing, roll-to-roll (R2R) processing, printed electronics, lab-on-a-chip, diagnostics

## Abstract

Microfluidic devices offer the potential to automate a wide variety of chemical and biological operations that are applicable for diagnostic and therapeutic operations with higher efficiency as well as higher repeatability and reproducibility. Polymer based microfluidic devices offer particular advantages including those of cost and biocompatibility. Here, we describe direct and replication approaches for manufacturing of polymer microfluidic devices. Replications approaches require fabrication of mould or master and we describe different methods of mould manufacture, including mechanical (micro-cutting; ultrasonic machining), energy-assisted methods (electrodischarge machining, micro-electrochemical machining, laser ablation, electron beam machining, focused ion beam (FIB) machining), traditional micro-electromechanical systems (MEMS) processes, as well as mould fabrication approaches for curved surfaces. The approaches for microfluidic device fabrications are described in terms of low volume production (casting, lamination, laser ablation, 3D printing) and high-volume production (hot embossing, injection moulding, and film or sheet operations).

## 1. Introduction

Microfluidic and micromachines have drawn significant attention since their introduction in the 1990s. The reduction in size, weight, and power consumption; improvement in sensitivity; and the characteristics of low-cost batch manufacturing of these devices have made the technology very appealing for numerous applications. Among various applications, microfluidic devices which handle small amounts of fluids for medical, biological, and chemistry applications are developing rapidly. These microfluidic devices have shown great potential to reduce cost in manufacturing, consumption of reagents, and time of analysis and to increase device efficiency and portability [1,2,3,4,5].

Microfluidics systems usually have channel dimensions of several tens to hundreds of microns and handle fluids in small quantities from 1 atto-litre to 1 nano-litre [3]. Early microfluidic devices typically employed silicon, quartz, or glass materials with well-established microfabrication photolithography, etching, and deposition processes [6,7,8,9,10]. Silicon patterning usually uses anisotropic wet etching by potassium hydroxide (KOH) and tetramethylammonium hydroxide (TMAH), or dry etching by reactive ion etching (RIE) and deep reactive ion etching (DRIE); quartz and glass patterning usually uses isotropic wet etching by hydrofluoric acid (HF). The traditional microfabrication approaches have several drawbacks:The cost of substrate per unit area is high. The cost of Corning Pyrex is in the range of 10–20 cents/cm^2^, whilst the cost of polymers such as polymethylmethacrylate (PMMA) is an order of magnitude lower in the range of 0.2–2 cents/cm^2^ [11].The process time is long and expensive. Iterative steps for cleaning, patterning, etching, and deposition, as well as reagents, are usually required, and all the processes are conducted in cleanroom environment.The finished profile is limited, with channels being shallow or circular, due to the characteristics of the etching process, whereas typically deep channels are preferred for most applications as they give higher surface areas and packing densities. High-aspect ratio silicon structures may be achieved with deep reactive ion etching (DRIE), but this is an expensive process.Some physical properties of traditional materials are not desirable. Biomolecules, for example, tend to adhere to a silicon surface. Prior chemical treatment may prevent the sticking problem, but the risk of contamination is increased. Silicon is also not an electrical insulator and can pose problems for electro-osmotic pumping. Moreover, silicon is also not transparent and therefore unsuitable for optical sensing. Quartz and glass require high temperatures and voltages for bonding, which may be problematic for device manufacturing.

Overall, the cost of production and some physical limitations of traditional micro-electromechanical systems (MEMS) materials pose difficulties for their wide use in microfluidic device applications. Much attention has focused on polymers as fabrication materials for microfluidic devices because of their unique characteristics:They are relatively cheap compared to silicon or glass in their unit area price. This characteristic is especially important for mass production or disposable usage in biomedical applications.They have a wide range of available material properties to fit with the requirements of processes and devices.They can offer good optical transparency, electrical insulation, or good gas permeability, which is important for cellular applications.The potential for the transfer of plastics processing knowledge from macro to microstructuring in the drive to mass produce microfluidic devices.

Before moving into specific manufacturing processes, we first consider the main polymer types used within microfluidic devices and their physical properties. We then describe the methods of manufacturing of a mould or a master that can be used for high-volume replication in terms of mechanical (micro-cutting, ultrasonic machining), energy-assisted (electrodischarge, micro-electrochemical, laser ablation, electron beam, focused ion beam), traditional MEMS and fabrication on curved surfaces. The processes are then arranged into low- and high-volume manufacturing techniques. The lower volume processes are typically serial, and we describe use of casting, laminate manufacturing, laser fabrication, and 3D printing. The high-volume processes—such as hot embossing, micro-injection moulding, and film or sheet processes—have a particularly important role for the commercial production of microfluidic devices.

### Polymer Types and Physical Properties

Polymers are repeating structural monomer units that are created through polyaddition (no loss of substance) or polycondensation (loss of water, alcohol, or other substance) reactions of the individual monomer units [12]. The most commonly used monomer chemistries include vinyl and acrylates, epoxy resins, thiol-enes, polyurethanes, and siloxanes. Vinyl monomers are those that contain carbon–carbon double bonds with the simplest vinyl polymer, polyethylene (PE), formed from the ethylene monomer. Replacement of one hydrogen atom of ethylene by methyl or chloride leads to propylene and vinyl chloride monomers, which are used to respectively form polypropylene (PP) and poly(vinyl chloride) (PVC) polymers. Polymerisation of the carbon–carbon double bond of the vinyl group is activated by a radical to turn the monomer into a radical and radical polymerisation to form the polymer. Vinyl chlorides and similar modifications are toxic and difficult to handle. In contrast, acrylates—a class of vinyl polymer from prop-2-enoyl (also known as acryloyl, acrylyl, or acryl) monomer—are generally easier to synthesise than pure vinyl polymers. It is relatively easy to produce monomers from the acryloyls. The most common starting point is acrylic acid and its methylated form methacrylic acid, the latter being used to produce the widely available poly(methyl methacrylate) (PMMA) (Figure 1). Acrylates are suitable for surface modification through chemical (activation by nitric acid) and physical surface treatment (plasma activation and corona discharge).

Epoxy resins are made using a two-step process of first making a diepoxy—typically from bisphenol A and epichlorohydrin—and then secondly crosslinking the epoxy groups. Compounds that add to the epoxy group include amines, amino acid, and thiols, all of which can initiate curing of the epoxy monomer. SU-8 is an epoxy-based negative photoresist and is one of the most commonly used materials for the development of microfluidic devices. In the case of negative photoresist, exposure to UV light hardens the material whilst the unexposed areas remain soluble and can be washed away during development (Figure 2A). The converse applies to a positive photoresist. Thiol-ene chemistry—based on the reaction of a thiol (comprising sulphur–hydrogen groups) and an alkene to form a thioether—has become important as material for microfluidic devices. The chemical resistance of thiol-ene polymers depends on the monomers used. It should also be noted that for microfluidics, unreacted thiol groups could lead to unspecific interactions with target molecules. The commonly used Norland optical adhesive (NOA) is a transparent UV curable thiol-ene photopolymer which interestingly can be made hydrophilic by oxygen plasma treatment. Polyurethanes (PU) are a further important class of polymers for the development of microfluidic devices and are created by the reaction of an isocyanate group with an alcohol or polyol. Both reversible and nonreversible channel creation can be achieved relatively easily by partial curing of the PU components. Siloxanes have an alternating silicon–oxygen polymer backbone. The most common amongst these is polydimethylsiloxane (PDMS), which has been widely used for creating microfluidic devices using a casting approach (Figure 2B).

Polymers are classified in terms of the chemistry of monomers (which link up to form polymers) but also a physical classification into three categories—thermosets, thermoplastic, and elastomers—which depends on the extent to which the polymers change shape when exposed to heat. If the polymer chains are highly interlinked (crosslinked) then the network is temperature-stable and the material is classified as a thermoset. The crosslinking can be initiated chemically—by addition of a curing agent with the monomer—or physically with either light or heat by addition of either a photoinitiator or thermoinitiator. At very high temperatures, thermosets will generally decompose but not melt since the glass transition temperature will be in the same range as the materials’ decomposition. Examples of thermosets include polyimide and SU-8. In the case of thermoplastics, the polymer chains are not crosslinked and so they are able to move inside the bulk, particularly at elevated temperatures. Thermoplastic polymers soften when the material is heated above the glass transition temperature—the point where polymers become rubber-like and are deformable—which makes them suitable to be shaped with injection moulding or hot embossing. For both of these processes, the polymer is first raised to elevated temperatures, and subsequently cooled down below the glass transition temperature to obtain a solid part. Thermoplastic polymers can be further classified as either crystalline or amorphous, with the former having a narrow glass transition range and the latter having a large range. Examples of thermoplastic polymers that are commonly used include PMMA, polycarbonate (PC), cycloolefin polymer (COP), and copolymers (COC). Elastomers are weakly crosslinked polymers which change their shape under external force and return to the original shape after the force is removed. In the case of elastomers, the glass transition is below the operating temperature and so they are rubber-like and can be deformed without excessive pressure, which makes them ideal for implementation of microvalves and micropumps. PDMS is one example of a frequently used elastomer.

Devices may be fabricated using either direct manufacturing or a replication approach; direct manufacturing is performed using typically either mechanical or energy-assisted methods. Mechanical methods are relatively traditional and use tools to remove unnecessary materials. For the energy-assisted methods, an energy beam of a certain wavelength interacts with the polymer and the effect can be constructive or destructive for polymerisation. Polymer molecules which loose crosslinking as a result of the interaction can be washed away or burned off; where crosslinks are formed as a result of the interaction, then micro- or nanostructures are formed. The beam scribes on polymer directly along pre-determined paths, or through a mask to induce the patterns for the structure. Lasers, electron beam (e-beam), and focused ion beam (FIB) are examples of this approach. For replication manufacturing, a mould with micro or nano features is first fabricated; replicates of this master are subsequently moulded. Typical replication methods include casting, hot embossing, and micro-injection moulding.

## 2. Mould or Master Manufacture

Fabrication of microstructures using machining approaches can be costly because of the capital costs of the equipment, but also the process can be time-consuming. The manufacturing cost can be greatly reduced if the processes in micro- or nanometres can be carried out using a replication approach. In this case, the micro- or nanostructures are fabricated only once for the masters or moulds of the process, and products can be duplicated from the masters. The masters have the inverted or the negative features of the device structure. There are requirements for their design as a result of the demoulding process. Importantly, the master and the moulded part are required to have a sufficiently low friction level between them so that the micro parts are not damaged or break during demoulding. Physically, the master is therefore required to have a smooth surface so that friction can be minimised. Chemical interactions between the materials of the master and the moulded device are also important, and surface treatment is sometimes used for control of these interactions. Geometrically, the master cannot have undercuts, otherwise it will interlock with the moulded part. A draft angle is normally used to reduce the demoulding force. Moreover, the master should have sufficient strength and hardness to maintain microstructures over the repeating moulding cycles.

Masters for replication usually consist of two major parts with different feature sizes. This is because a microfluidic device usually consists of micro- or nano-scale functioning structures and macro interfacial structures with the outside world. Here, we will refer to the two parts as the tool and the mould insert [13]. The tool provides the structural support and the necessary vacuum and heating/cooling systems for the polymer device and the moulding processing; the tool is usually fabricated with classical machining method. The mould insert provides the micro/nano features of the polymer device and can be fabricated with various methods which are subsequently described further. There are usually multiple mould inserts for a process, and they are interchangeable. The inter-changeability lowers the total cost of the master manufacturing since trial-and-error is usually required to achieve the optimised design of a mould [14] and only the inserts are to be re-fabricated, not the whole mould. Here, we focus on the fabrication of the mould insert, since they determine the final features and functions of the microfluidic devices.

Conventional microfabrication, such as lithography and wet/dry etching, is the natural choice when mould inserts with micro or nano structures are required and where the materials are usually limited to silicon, glass, or polymers, particularly photoresists. With miniaturisation and advances in tooling, traditional machining in micro-size, or micro engineering technology, has achieved some fine structures [15,16,17,18,19,20,21]; metals can also be used as the master material. Therefore the feasible ways to fabricate moulds include both the traditional MEMS approaches and micro-engineering [17]. The currently available master-making methods are diverse, the most commonly used methods are described below; more detailed descriptions are provided elsewhere [15,17,21,22,23,24,25]. For ease of discussion, we categorised the methods as mechanical, energy-assisted, and MEMS-based methods [17].

Mechanical methods remove any unwanted portion of the master by mechanical forces with sharp tools or particles upon contact with the master in a small area. The tools or particles result in high stress locally and causes plastic or brittle breakage of the master materials. The tooling for these processes must be stronger than those of the master materials. Microcutting with computer numerical control (CNC) machines and ultrasonic micromachining fall into this category. For the energy-assisted method, concentrated energy is applied to the master material either through electric potential, laser, ion, or electron beams. Electrodischarge machining (EDM), electrochemical machining (ECM), laser ablation, microstereolithography (μSL), focused ion beam (FIB), and electron beam (e-beam) machining can be grouped in this category. Table 1 summarises the achievable capabilities of the methods that have been mentioned and is compiled using various sources [11,15,16,17,18,19,20,21,24,26,27,28,29,30,31,32,33]. It must be recognised that these are evolving areas and that advances within the different technology areas will lead to improved attributed and figures of merit. Most of these methods can also be used to fabricate polymer parts directly. Due to the serial-processing characteristics of direct fabrication, however, it is not economic to adapt this route for volume manufacture, and thus these methods are mostly used for fabricating masters [34].

### 2.1. Mechanical Methods

#### 2.1.1. Micro-Cutting

Micro-cutting includes drilling and milling, which uses drill bits less than 3 mm in diameter and rotating at much higher speeds (up to 180,000 rpm) than the traditional methods. The micro-tools are typically tungsten carbide with coatings such as titanium nitride (TiN), titanium–aluminium nitride (TiAlN), or diamond. The micro-tools are usually fabricated by other methods of smaller resolution, such as FIB or grinding. Drill diameters less than 25 μm have been demonstrated [35]. Diamond or diamond-coated tools are used for high-surface finish on soft materials and cubic boron nitride (CBN) for hard materials.

Usually, the tool speed and position are controlled by CNC, where a computer sends instructions to the motor and servos. The method can also use files from computer-aided design (CAD) or computer-aided manufacturing (CAM) directly, and the manufacturing is convenient and fast. Thermal expansion during machining is less of a problem in micro-milling when compared with traditional methods since the larger surface-to-volume ratio in the micro-scale aids cooling. High speed milling strategies have been developed that allow the tool to stay in almost constant contact with the workpiece to maximise the material removal rate, and hence micro milling has relatively high removal rates compared to other methods [17]. Channels, through holes, and chambers are frequent examples of the application of micro-milling [36,37,38].

Micro-milling has many appealing characteristics—it can easily manufacture 3D structures of high aspect ratios with inclined angles on side walls, which have beneficial importance for demoulding [13,15]. There also exists a wide array of machinable materials for micro-milling—metals, polymers, composites, or ceramics are all acceptable. Stainless steel, in particular, is advantageous as it provides good wear resistance and master lifetime [11]; the stainless-steel master cannot be machined with traditional MEMS approaches. Moreover, the turnaround time is shorter than for the MEMS process, as there is no mask fabrication and lithography involved; the simplicity is quite attractive for prototyping [39]. Because the tool is in contact with the workpiece during the machining process, the location of the machining surface can be easily tracked, in contrast to other methods such as ultrasonic machining or ECM where the tool is not in contact with the workpiece.

The major limitation of micro-cutting is the relatively poor minimum feature size and surface finish that can be achieved compared to other methods; sharp corners are also difficult to obtain [40]. The elastic deformation of the tool and the workpiece, vibration of the machine, thermal deformation, and the tool itself all can affect the accuracy of the machining [22]. Furthermore, very hard or brittle materials can present difficulties. A diamond-coated tool to feed with small depth and slow speed can handle the difficulties, but the wear rate of tools can be high [22]. It is also difficult to predict or detect breakages of the tool because of the small size. Additional systems are needed for tool-condition monitoring such as laser tool measuring or tool-workpiece voltage monitoring, and routine off-line tool checks are needed to solve the problem [41]. Micro-milling can, however, be a very effective and low-cost way of prototyping microfluidic devices. Chiriaco et al. have described a convenient fabrication approach for multilevel/3D PMMA fluidic mixer device comprising three layers structured using micro-milling and with a through hole to connect the different layers which were joined by spin-coating hot isopropyl alcohol on the surface of the substrate [42]. The hydrophilicity of the PMMA channels was improved by O_2_ plasma treatment.

#### 2.1.2. Ultrasonic Machining

Ultrasonic machining (USM) removes the surface of workpiece with abrasive particles vibrating at ultrasonic frequency. The abrasive particles are hard materials such as boron carbide, aluminium oxide, and silicon carbide, which are usually carried by water or oil in a slurry state. The sonotrode of the tool, which causes vibration of the abrasive particles, is brought to the vicinity or machining area of the workpiece, which is covered by the slurry. The sonotrode vibrates the slurry particles and the impact of the abrasive particles ablates the surface of the workpiece. The slurry is injected between the sonotrode and the workpiece constantly to replace worn-out abrasive and to carry away workpiece debris and heat generated from the impact. The breakage of workpiece generally arises from direct impact or cavitation erosion effects. For the direct impact effect, the workpiece is abraded upon direct pounding by the tool through the abrasive particles caught in between the tool and the workpiece. For the cavitation erosion effect, the burst of the cavitation bubbles in the slurry sends shockwaves to abrasive particles and in turn impacts the workpiece. The size of the abrasive particles and the amplitude of the vibration have major influences on the accuracy and surface roughness of the machining; tool wear, workpiece materials, and depth of machining also have effects on accuracy. The method is also called ultrasonic impact grinding.

Ultrasonic machining does not contact the workpiece directly for material removal, and thus little residual stress is generated on the workpiece. Because of the indirect contact, the thermal effect on the workpiece during machining is also small. Brittle materials, such as ceramics, silicon, and glass, are generally easier to machine because of the brittle-breakage principle of the method; soft or ductile materials are less efficient because much of the vibration is absorbed elastically or the workpiece is not chipped away easily due to ductility. The operating cost of the machine is low and the skill requirement for operation is relatively modest [25].

The major limitation of USM includes low material removal rate [43], high tool wear ratio [15], and accuracy problems. The high tool wear results from the abrasive effect on both the tool and the workpiece, and thus deep holes are difficult to machine; constant tool change might be required for prolonged machining. Sintered diamond tools seem to provide a solution with a lower wear. The vibrational characteristic of the tool cause problems in tool loosing and machining accuracy. The on-the-machine tool preparation with tool integrated with the machine and the vibration of the workpiece, instead of the tool, improve the accuracy of the method [21,22].

### 2.2. Energy-Assisted Methods

#### 2.2.1. Electrodischarge Machining

Electro-discharge machining (EDM) removes materials through a spark erosion process [16]. The workpiece and the tool are both electrically conductive and are both soaked in a dielectric fluid and placed close to each other. A voltage pulse is applied across the workpiece and tool; the small distance between the two results in a high electric field and causes electric breakdown of the dielectric fluid. The arc from the electric breakdown melts the workpiece locally and achieves the machining purpose. The dielectric fluid is constantly flushed to carry away the burned material and heat generated from the process. The flushing process is important because the burned material re-solidifies in fluid as particles and can quickly accumulate; the build-up of particles can cause electrical short and stop the machining. Important parameters for EDM generally include pulse energy, pulse rate, polarity, material and geometry of the tool, type of dielectric fluid, gap between the tool and the workpiece, and the flushing of the dielectric fluid. Micro-structures such as holes, grooves, cavity, convex, and rods are among the possible applications [22].

The advantage of EDM lies in the wide choices of workpiece materials and geometry [11]. Hardness of the workpiece material is not a concern for this process [21] because the materials are essentially burned away with high temperature and thus stainless-steel can be used as the master material. The machining forces and the residual stress onto the workpiece are also negligible because of the non-contact machining.

The limitations of EDM are the machining speed, accuracy, finish surface, and dielectric fluid flushing. The machining speed of EDM is relatively slow [15]. Deionised water has been used to improve machining speed, but it reduces the machining accuracy due to a widening gap from the electrode [22]. Tool wear is difficult to predict, which results in difficulty in identifying the location of the tool. A system to regenerate the tool in situ has been demonstrated with side-by-side ECM by Layouni et al. [44]. To identify the exact shape of the tool in situ and thus its exact location, however, can still remain a challenge. The process of EDM usually leaves a layer of melted and re-solidified material (recast) at the machining zone, which tends to be hard and brittle with decreased fatigue strength. Further machining to remove this layer might be necessary for long-term repeating applications. The flushing of the dielectric fluid can be difficult, especially when the feature is deep and the gap between the tool and the workpiece is only several microns. The tool can be oscillated to improve the flushing efficiency in some machines, but accuracy may be reduced as a result.

#### 2.2.2. Micro-Electrochemical Machining

Electrochemical machining (ECM) utilises the electrochemical dissolution of workpiece in electrolyte upon application of electric potential. The setup of ECM is similar to that of electroplating, but with the workpiece on the anode and the tool, or the electrode, on the cathode. To localise the dissolution for machining purposes, one shapes the tool or electrode to achieve inhomogeneous current density; preferential dissolution rate around the tool can be therefore obtained and controlled machining is achieved. The different dissolution can also be achieved by the gap between the tool and the workpiece, partial insulation of the tool or workpiece, high current density, and voltage pulse duration [18,19]. Suitable electrolytes are chosen for the workpiece material and are constantly injected in between the tool and the workpiece; this practice avoids plating and changing the shape of the tool, as well as carrying away the heat from the process. The flush of electrolyte also prevents build-up of debris or gas in the gap, which can slow down the dissolution due to density change or short circuit of the process.

Electrochemical micromachining (EMM) is a variant of ECM at the micro-scale. Regular lithography on workpiece has helped to achieve differential dissolution and to enable micro patterning with ECM [45,46]. Undercuts, however, can be a concern for machining accuracy. Further development has shown promising machining tolerance by applying ultra-short pulses in the nanosecond duration [47]. Important parameters of this process include the electric pulse condition (voltage, duration), workpiece-tool gap, choice of electrolyte, and electrode size. Usually, a higher pulse voltage is used for rough machining, and a lower one is used for finish machining. Similar to EDM, the workpiece material has to be conductive, and hardness of the material is immaterial to the process. Cavities, holes, channels, and 3D structures are common applications (see Datta et al. [48], Landolt et al. [49], and Ehrfeld [50]).

Electrochemical machining is relatively easy for design and setup because of the CNC capability. The running costs are relatively low and the skill demand of labour is not high [25]. The wide range of materials it can process is also advantageous. The machining is non-contact and dissolution-based, and thus no residual mechanical or thermal effects are presented on the workpiece surface [22]. The high surface finish achievable by this method is ideal for master making because it significantly reduces the friction during the demoulding process. Some disadvantages of ECM are similar to those of EDM. The removal rate is small; therefore, the processing time can be long. The machining accuracy also needs consideration because the tool does not contact with the workpiece directly. The flow pattern and temperature of the electrolyte can also affect the accuracy [22]. The initial investment of an ECM machine is expensive [25], which can impede the usage of the method.

#### 2.2.3. Laser Ablation

Excimer micromachining relies on the interaction between UV pulsed laser radiation and the material to be machined. The short wavelength means that radiation is efficiently absorbed in the surface layers of all but a few materials; the short pulse duration ensures high peak absorbed power densities. The combination of UV and short pulses results in the removal of surface layers by one of a number of mechanisms—vaporisation in the case of metals and ceramics, molecular disintegration (photoablative decomposition) for many polymers at less than 300 nm, or interface effects (stripping by exfoliation of thin films up to a few micrometres thick). Successive laser pulses cause further material ablation, and thus controlled localised milling into the part, which constitutes the basic act of machining. Energy densities are typically in the range of 1–10 J/cm^2^ at repetition rates up to a few kHz; machining takes place at rates on the order of tenths of micrometres per laser shot.

When the laser beam strikes the workpiece surface, reflection, absorption, and conduction of the beam occur. When the workpiece material absorbs strongly in the incident wavelength, the absorption of radiation causes breakage of chemical bonds or vaporises the material. The higher the absorption rate, the more efficient the ablation. To increase the absorption and reduce the reflection, one can utilise methods such as changing surface finish, applying surface coating, and oxidising the workpiece surface [24]. In the case of plastics, excimer lasers are the most efficient because the photon energy level is similar to that of the molecular bonds for plastics [22]. In the case of metals, oxygen is usually utilised to assist the machining capability, because metals have higher reflection and thermal conduction which tend to dissipate incident energy. The gas-assisted reaction is similar to the process of oxygen-acetylene torch cutting.

The pulse length of laser predominantly determines the characteristics the outcome of the machining. Lasers of ultra-short pulse in the femtosecond and picosecond delivers higher energy, and sublimation of the workpiece material happens at the point of focus. The heating of the workpiece is localised, and there is only negligible heat-affected zone (HAZ) in the workpiece. Lasers with longer pulse in the nanosecond and microsecond deliver lower energy. There is enough time for the laser energy to be absorbed in the workpiece and the thermal wave propagates into the material. Melting and re-solidification, or recast, occur, and HAZ, micro cracks, shockwave surface damage, and debris are also possible side effects under this case [17].

The minimum lateral size of laser ablation is primarily associated with the optics and the wavelength of light (Table 2); a smaller wavelength generally creates finer features. The beam power and quality also affect the minimum feature size. The parameters controlled in this method are wavelength, power, pulse duration, and pulse repetition rate of the laser [15]. Lasers can be used for fabrication of moulds that can then be used for higher volume replication of microfluidic devices. A femtosecond laser has been used for fabrication of metallic inserts for microinjection moulding [51].

The advantage of laser ablation includes its wide acceptance of workpiece materials. Metals, polymers, ceramics, composites, semiconductors, diamond, graphite, and glass are all machinable [21], even those not feasible in MEMS processes. The force involved is also much smaller than that in a mechanical process. The workpiece may be thin and elastic, and requires no mechanical strength [22]. It is also a non-contact machining without mechanical wear and may work well in atmosphere environment. The potential limitations in the use of lasers includes the non-uniform depth of cavity and slanted side walls that are produced [22]; multiple shots are required to reach the target depth, and small variation exists in individual shots. Small feature tolerance is generally difficult to achieve because of the size of focus, and the outline of the beam is not exactly clear [22]. The material removal rate is usually small, and thus, consequently, the process is slow and the equipment is relatively costly [21,22].

Excimer beams have a typically broad spatial profile and poorly defined mode structure. Unlike other lasers, focal point applications are rare; most processing is performed using projection optics, where the beam is used to illuminate a mask, whose de-magnified image is then focussed on the part. The mask may define a simple motif, e.g., circle, slit, or more complex motifs which are then projected as a whole onto the part, e.g., alphanumeric characters. The final machining pattern on the part may be built up from repetition of selected motifs associated with part motion in X, Y, and laser firing; machining depth is mainly controlled by local shot dose. Fixed masks are typically made of laser cut or chemically etched shimstock; multiple masks may be mounted on a motorised selector carousel. Dynamic masks use, for example, a motorised slit or rectangular variable aperture (RVA). The choice of mask type is determined by processing requirements. Excimer processing, particularly of polymers, often benefits from a shield gas (typically He) during processing, whilst process debris/fumes need to be safely extracted from the process area and taken to a suitable exhaust.

#### 2.2.4. Electron Beam Machining

Electron beam (e-beam) machining utilises the impact of incoming electrons to machine the workpiece. The machine usually consists of an electron generating source such a thermoionic tungsten filament or field emission gun (FEG), an electrostatic or magnetic optics to focus the e-beam, a workpiece carrier, and a vacuum chamber. A vacuum environment is required to minimise electron scattering that occurs in an atmospheric environment. Electrons from the electron gun are accelerated and focused through electrostatic or magnetic optics and bombard the workpiece at high speed. The workpiece material is melted or vaporised upon the thermal energy resulted from the electron bombardment, such as in welding or annealing applications; e-beam melting of metal powders in rapid prototyping is one example. E-beams can also change the molecular chains or chemical bonding of the workpiece material, such as in the lithographic applications; photoresists change their selective solubility in developer solvent after being exposed to the e-beam. The energy level of the e-beam determines the change of the workpiece material.

Lithography is the most general usage of e-beam machining in fabricating micro parts. Melting by e-beam is also used in physical vapour deposition (PVD) or in the application of making micro-masters [52]. A modified SEM is the typical practice for many research laboratories to conduct e-beam lithography; the machine usually is equipped with a pattern generator and a beam blanker, which redirects the e-beam to prevent some areas from being exposed to the electron bombardment. The feature resolution depends not only on the size of the e-beam, but also on the scattering of the electrons after impact, because the scattering contributes to further exposure of the photoresist. The resolution is also constrained by the stability of the machine. A high resolution of 10 nm or less can be achieved by e-beam lithography. E-beam can also initiate deposition with precursor gases. Tungsten nanowire with diameter about 70 nm has been demonstrated by Klein et al. [53].

The advantage of e-beam machining is its high resolution; features in the nanometre range can be easily generated. The method is non-contact, and thus there are little mechanical or thermal residual stresses. The method can also conduct direct patterning without the need of a mask. It is therefore a primary means of making photomasks for lithographic applications. The main limitation of e-beam machining is the process time. Many hours are usually required to expose a 4” wafer area, and the need of a vacuum chamber limits the size of the workpiece; moreover, the initial capital investment required for an e-beam machine is high.

#### 2.2.5. Focused Ion Beam Machining

Focused ion beam (FIB) machining utilises ions with high kinetic energy to remove or add material by momentum transfer. Ions from an ion source (usually liquid metal ion source or LMIS, typically Gallium) are accelerated and impinge onto the surface of workpiece or gas introduced to the process. As ions are far heavier than electrons, ion beams bombard the target with greater energy and the scattering is relatively small compared to that of an electron beam. For the subtractive process, ions sputter away the workpiece material directly by bombardment, or initiate the etching process of the added gas on the workpiece; the energised gas molecules react with workpiece material to form volatile substance and removed by vacuum [54]. For the additive process, a gas injection system (GIS) is used and a precursor gas is injected close to the surface of the substrate (typically 100 µm). Having the injector needle close to the sample surface allows the gas flow to be kept to a minimum, which ensures there is little disruption to the system’s vacuum; then, the effect of the gas is localised. The gas is adsorbed onto the surface of the workpiece and is decomposed by the ion bombardment to form a non-volatile substance on the workpiece surface through the process that is the same as chemical vapour deposition. The volatile by-product is removed by vacuum [54]. Carbon may be deposited as a protective or capping layer, whilst metals such as W, Au, Al, Pt, and Co and inorganic materials such as SiO_2_ may also be deposited this way.

The FIB can engrave or deposit patterns on the workpiece directly from a CAD file. The direct-write processes include milling, implantation, deposition, and etching [30]. The method can also be used through a stencil mask and projected onto the workpiece to create the desired pattern. A spot size of 5 nm is feasible with a FIB [54]. Applications of FIB have been centred on mask repairing, device modification, circuit debugging, online inspection, and fabricating nanostructures or lenses. All processes are conducted in a vacuum chamber. For the sputtering operation, factors such as ion energy, ion species, incident angle of the beam, and surface bonding energy of the workpiece material are all important for the removal rate of the workpiece [30]. The very high resolution is the major benefit of FIB, and it can work on all kinds of materials, including metals, inorganic semiconductors, and ceramics. The limit, as with the e-beam, is the extremely low processing rate. A FIB has to be operated in a vacuum environment and may only pattern a small area, for example 250 × 250 μm.

Most widespread are instruments using liquid metal ion sources (LMIS), especially gallium ion sources. Ion sources based on elemental gold and iridium are also available. In a gallium LMIS, gallium metal is placed in contact with a tungsten needle, and heated gallium wets the tungsten and flows to the tip of the needle where the opposing forces of surface tension and electric field form the gallium into a cusp shaped tip called a Taylor cone. The tip radius of this cone is extremely small (≈2 nm). The huge electric field at this small tip (greater than 108 volts per centimetre) causes ionisation and field emission of the gallium atoms. Source ions are then generally accelerated to an energy of 1–50 keV (kiloelectronvolts) and focused onto the sample by electrostatic lenses. LMIS produce high current density ion beams with very small energy spread. A modern FIB can deliver tens of nanoamperes of current to a sample or can image the sample with a spot size on the order of a few nanometres.

Unlike an electron microscope, an FIB is inherently destructive to the specimen. When the high-energy gallium ions strike the sample, they will sputter atoms from the surface. Gallium atoms will also be implanted into the top few nanometres of the surface, and the surface will be made amorphous. Because of the sputtering capability, the FIB is used as a micro- and nanomachining tool in order to modify or machine materials at the micro- and nanoscale. FIB micromachining has become a broad field of its own, but nanomachining with FIB is a field that is still developing. Commonly, the smallest beam size for imaging is 2.5–6 nm. The smallest milled features are somewhat larger (10–15 nm), as this is dependent on the total beam size and interactions with the sample being milled.

FIB tools are designed to etch or machine surfaces; an ideal FIB could machine away one atom layer without any disruption of the atoms in the next layer, or any residual disruptions above the surface. Yet, due to the sputtering, the machining currently typically roughens surfaces at the submicron length scales. A FIB can also be used to deposit material via ion beam-induced deposition. FIB-assisted chemical vapour deposition occurs when a gas, such as tungsten hexacarbonyl (W(CO)_6_), is introduced to the vacuum chamber and allowed to chemisorb onto the sample. By scanning an area with the beam, the precursor gas is decomposed into volatile and non-volatile components; the non-volatile component, such as tungsten, remains on the surface as a deposition. This is useful, as the deposited metal may be used as a sacrificial layer in order to protect the underlying sample from the destructive sputtering of the beam.

At lower beam currents, FIB imaging resolution begins to rival the more familiar scanning electron microscope (SEM) in terms of imaging topography; however, the FIB’s two imaging modes, using secondary electrons and secondary ions, both produced by the primary ion beam, offer many advantages over SEM. FIB secondary electron images show intense grain orientation contrast. As a result, grain morphology can be readily imaged without resorting to chemical etching. Grain boundary contrast can also be enhanced through careful selection of imaging parameters. FIB secondary ion images also reveal chemical differences, and are especially useful in corrosion studies, as secondary ion yields of metals can increase by three orders of magnitude in the presence of oxygen, clearly revealing the presence of corrosion. A method of 3D imaging is to serially slice the component and image each section in 2D—the resulting array of 2D images are stacked to form a 3D model of the part. This is a destructive method, but similar to magnetic resonance imaging used in medical scanning.

### 2.3. Traditional MEMS Process

Lithographic methods utilise techniques frequently used in microfabrication with photolithography, wet (KOH, TMAH, etc.), and dry (RIE and DRIE) etching, as well as FIB and e-beam. Many processes are carried out on silicon wafers. Silicon masters have excellent rigidity and wearability, and can achieve small feature size. The material, however, is brittle, and the manufacturing is time-consuming. MEMS processes along this line can be found in numerous books and are not elaborated here. Polymers, on the other hand, can be patterned with simple lithographic procedures; therefore, they are attractive. Polymeric materials in focus are SU-8 and PMMA because they make thick and high-aspect-ratio structures; this feature is in line with the 3D characteristics of masters for moulding.

SU-8 can achieve structures with thickness more than 1 millimetre and aspect ratio around 40:1 [32]. Standard equipment with UV light between 320 and 450 nm for contact lithography can be used for the lithography. The developed structure usually has smooth surfaces. Simplicity and compatibility with standard cleanroom environment are the benefits of an SU-8 UV approach. SU-8 has been used as masters for PDMS in microfluidics (Figure 2B) or optics, or directly as structures in many occasions [55,56,57,58,59,60]. Abgrall et al. [61] and del Campo et al. [62] have informational review on SU-8. For structures with higher aspect ratios, for example 100:1, X-ray lithography of shorter wavelength is preferred. Highly collimated X-rays with small scattering can achieve resolution in the 50 nm range. The achievable smooth and vertical side walls are also the benefits of this method. The primary problem with X-ray lithography is the requirement of a synchrotron as the light source, which is expensive and not available in many facilities. PMMA is commonly used as the photoresist.

Masters for replication created by the above methods are usually further processed to increase their lifetime or to provide a suitable interface of the mould with process polymers (e.g., many polymers tend to stick to silicon); electroplating is a common practice. Nickel and its alloys, e.g., NiCo or NiFe, or copper are commonly used as electroplating materials. The process of combining X-ray lithography, electroplating, and moulding is termed as LIGA (Lithographie, Galvanoformung, Abformung from German). The LIGA processes can make high-aspect ratio electroplated moulds and have good surface smoothness. The drawbacks of the electroplating process are the slow deposition rate and high residual stress inside the deposited layers, which can deform the master mould. Variant approach replacing the X-ray lithography with a UV one is also widely used and usually termed as UV-LIGA or modified LIGA. UV-LIGA has the benefit of wider equipment availability at the cost of reduced achievable aspect ratio. For more details on LIGA process, readers can refer to the following references for more information [63,64].

### 2.4. Mould Fabrication on Curved Surfaces

Previous discussions usually focus on masters of planar feature faces. For continuous manufacturing, however, reels or rollers are usually used for printing and packaging purposes. Therefore, masters of curved or circular surfaces are equally important. The transfer of patterns onto a roller is the main issue of the discussion in this section. The regular photolithographic techniques on a planar surface face difficulty with curved substrates because the photoresist cannot be coated evenly onto the roller and the reflection of ultraviolet (UV) light from curved surfaces can cause uncertainty in absorption dosage and pattern distortion.

Various approaches have been demonstrated to create patterns on curved or nonplanar surfaces. Direct methods use electron beam, ion beam, or laser to create pattern on photoresists or substrates directly [65,66,67,68,69,70,71,72,73]. These approaches do not require a mask to generate necessary patterns, and thus features can usually be transferred without difficulties. Because of the serial characteristic of these processes, however, the size of the mask and throughput can be constrained, and the equipment is also usually expensive. Other than direct methods, modified LIGA processes [74,75] use flexible covering as pattern and variants of lithographic methods [76,77] adapt intermittent UV exposure to pattern periodic features on roller surfaces. Dedicated equipment is less demanding in these methods. The achievable pattern resolution, however, is limited; continuous patterns on the surface can also be difficult due to intermittent exposure [78]. Moreover, soft-lithographic techniques with PDMS or other polymers [79,80,81,82,83,84,85] provide relatively cheap, easy, and direct methods to prepare elastomer patterns or moulds. These patterns can achieve resolutions in the nanometre range and are applicable to a variety of substrates. These techniques, however, have difficulties in the alignment of layers or possible pattern deformation under strain during application. Other methods use combination of various approaches [78,86,87], and many of these techniques are adapted for pattern creation on rollers.

In line with the above techniques, there are two major approaches to fix patterns onto rollers. One is to engrave, etch, or grow the pattern on the roller directly; the other is to create the pattern on a separate flexible media first, and subsequently to attach the media onto the roller. A variety of different approaches have been demonstrated. For patterning onto the roller directly, Huang et al. [76] and Wu et al. [77] used a modified LIGA process with dip coating to coat photoresist onto the roller, exposed with stepped-rotating lithography, and electroless nickel plating on an aluminium roller. Jiang et al. [88] attached a dry film resist to the roller surface, exposed with a flexible mask, and wet-etched to create micro grooves on a copper roller. Marques et al. [75] used a patterned PMMA template to wrap around a cylinder and electroplated with nickel. Because the pattern is an integral part of the roller, the approach fabricates durable rollers which can sustain higher pressure and temperature, as well as solving the possible mould-sliding problem in the mould-wrapping approach. A uniform coating of photoresist or a perfect wrapping of dry film on the roller can, however, be difficult [76]. The UV exposure facility also needs modification for patterning on circumference. Resolution can be limited by the optical setup. A PDMS soft mould on the roller cast by wrapping of PC film is also demonstrated [89,90]. A soft mould provides good conformal contact with the substrates. The strength and temperature durability, however, are not as good as the case for a metal mould.

A thin electroplated nickel mould is the most commonly used method for ease of mould attaching [67,91,92,93,94,95,96]. Most nickel moulds have a thickness of around 50–100 μm and are attached to the roller by taping, double taping, or clamping. Chang et al. [91] electroplated a thin nickel mould against a PC film, and wrapped the film with a cushion layer onto a steel cylinder. A number of other similar approaches have been demonstrated [93,94,96]. Velten et al. [97] used a silicon film, of thickness about 40 μm achieved by dicing-and-thinning process, which was wrapped around a steel cylinder. The feature size on the silicon film was in the range of 100 nm originally achieved by dry etching. Shan et al. [98] used a copper/liquid-crystal polymer (LCP)/copper laminate as the flexible mould to wrap around a steel roller; the copper layers are used as both the pattern and backing layers. These techniques allow the mould to be fabricated with an ordinary LIGA process on planar surfaces and are relatively simple and fast. The adhesion between the mould and the roller under elevated temperature and pressure during embossing is, however, not strong, and gaps can exist between the contact of the mould and the roller; sliding, warping, and distortion of the mould are concerns for this approach. Shan et al. [99] demonstrated an alternative approach by using a planar rigid nickel mould and a supporting plate sandwiching the embossing material, passing through rollers. This alternative is easy for mould mounting and exchanging, and imposes no restriction on mould thickness for flexibility. The alternative, however, cannot pattern continuously due to size limit of the planar mould, and will be difficult to integrate with other roller operation.

## 3. Low-Volume Production

### 3.1. Casting

Casting is the fabrication process primarily for silicone-based elastomers to mould or to be used as a stamp to create micro- or nanostructures, and is often referred to as soft lithography. There are a number of variants of soft lithography, including microcontact printing (μCP), replica moulding (REM), microtransfer moulding (μTM), micromoulding in capillaries (MIMIC), solvent-assisted micromoulding (SAMIN), as well as phase-shift photolithography and cast moulding [83]. The techniques aim to reproduce features smaller than the barrier of optical lithography (≈100 nm), and have seen proliferation for small-scale prototyping because of their low capital cost, simple procedure, high fidelity, and advantageous material properties and chemistries [83,100]. Numerous examples have demonstrated the capability to produce features in several tens of nanometres, although more common applications have features at a few microns. Most publications used PDMS or silicone rubbers but other elastomers such as polyurethanes [101,102] and polyimides [103] have also been investigated. For simplicity, the discussion here will be limited to PDMS.

Several mechanical, chemical, and optical properties of PDMS make it appealing for microfluidic applications. According to McDonald et al. [1] and Xia et al. [83], the material can be deformed under external force, and thus non-planar surfaces can also be replicated easily. The material is durable to be used as a stamp for a period of time without noticeable wear. The surface can also be changed easily by oxygen plasma treatment to form an active self-assembled monolayer which makes it easy to bond with a wide range of materials, such as glass, silicon, quartz, PDMS, polyethylene, and polystyrene [104], to form closed structures. Moreover, the polymer has good chemical inertness, and thus most molecules or polymers will not stick permanently to its surface. The material is nontoxic, with good gas permeability, allowing cell culturing, and the material cures at low temperatures. Finally, the fabricated elastomer is transparent down to 280 nm for optical observation. PDMS does, however, have several drawbacks for particular applications. First, the material swells if in contact with several nonpolar organic solvents, such as diisopropylamine, triethylamine, pentane, and xylenes [105], presenting difficulty for maintaining feature sizes. Second, the softness and thermal expansion make it difficult for PDMS to achieve high accuracy for a large area; PDMS microstructures also tends to collapse with high aspect ratios, although design rules allow these issues to be ameliorated [83] and use of an SU-8 mould can lead to higher aspect ratio features. Overall, PDMS is especially useful for cellular and protein applications [106,107,108,109,110,111], patterning of biological and non-biological materials [112,113,114], and sensing components [115,116].

PDMS casting on an SU-8 master mould is a relatively low cost process for the development of microfluidic devices (Figure 2). The procedure therefore starts with fabrication of the master mould by spin coating negative SU-8 resist onto the substrate and then a soft (first) bake by passing through a temperature cycle which is dependent on the SU-8 resist thickness. High temperatures for the soft bake need to be avoided to prevent thermal activation prior to UV exposure. The pattern from a photomask is transferred onto the SU-8-coated substrate by exposing to UV light and then a post exposure bake to accelerate the SU-8 polymerisation. The SU-8 is then developed to give the designed microfluidic architecture. PDMS casting on the fabricated SU-8 master mould is carried out by first preparing the PDMS liquid polymer—mixing base elastomer and curing agent or catalyst—and pouring the mixture onto the SU-8 mould. Curing of the PDMS is carried out either at room or at elevated temperature (typically 40 to 70 °C) for polymeric crosslinking, and then peeling off the elastomer for demoulding. Degassing of the mixture during mixing and moulding is usually necessary to eliminate cavities or bubbles in the moulded structure. The PDMS polymer with microfluidic architecture can be sealed by joining to a further substrate by oxygen plasma [117] or corona discharge [118], and this can be carried out outside of a cleanroom environment.

PDMS casting on an SU-8 mould can be used for the development of microfluidic devices with multiple layers, which allows for the development of Quake-type valves [119] and also combining a series of valves to create a pumping system. Microfluidic devices with multiple layers can allow for more complex fluidic operations which could be necessary for point-of-care diagnostic or other applications. For multi-layer devices, it can, however, be difficult for the different layers to achieve manual alignment and secure bonding. Moreover, the world-to-chip interfaces for cast microfluidic devices is not straightforward.

We have previously described addressing a 24 element array of microchambers through a series of row and columnar pneumatically actuated normally closed (NC) valves [120]. A three-layer structure comprising a fluidic, flexible PDMS membrane and control layers (Figure 3) was used, wherein the fluidic and control layers were cast in PDMS from an SU-8 master mould that was made using lithographic procedures. Applying a vacuum within the control channels of either the row or columnar valves will lead to deflection of the flexible PDMS membrane and allow fluid to flow respectively through the row or columnar channels.

### 3.2. Laminate Manufacturing

Within laminate manufacturing, individual layers are first structured and then joined together in a stack to create microfluidic devices for a variety of applications [121]. We refer here to laminate manufacturing that typically involves 4 to 12 layers and not to the sheet and roll-to-roll processes (described in Section 4) which can be used for higher volume manufacture. Laminate manufacturing allows a wide variety of joining approaches and consequently an extensive choice of materials for the individual layers—including polymers (typically polycarbonate, PMMA, COC), adhesive tape, and glass—which can offer specific benefits such as optical clarity or biocompatibility [122]. Structuring of the individual layers is carried out by first designing within CAD and then cutting with either a knife plotter (xurography) [123,124] or a laser, with the knife plotter being lower cost and easier to use but with lower resolution. Joining of the layers creates the channel—with the channel height being controlled through the choice of the individual layer thickness—and can be carried out using either adhesive, thermal, or chemical methods. Adhesive transfer tape has been widely used for prototyping microfluidic devices with a variety of functions including mixing, particle separation, and high temperature reactions [125,126]. PCR adhesive tape can offer easy, low-cost, and reversible bonding with manual pressing and no requirement for heating [127]. Kang et al. have reported the use of (3-glycidyloxypropyl)trimethoxysilane (GLYMO) for bonding poly(methyl methacrylate) PMMA to polyethylene terephthalate (PETE) track-etched membranes [128].

Laminated microfluidic devices—having channel features above 100 μm and thicknesses ranging 2.4 to 5.2 mm—have been described [129] using PMMA sheets with individual thicknesses between 0.2 and 2 mm that were structured using a CO_2_ laser (Epilog Mini 18, Epilog, USA) (Figure 4). The bonding was carried out by spreading small amounts of ethanol and the different layers aligned using custom holders with pins for alignment of the layers. A custom heating plate and manual press were used for bonding with temperature of 70 °C and pressures (≈1.6 MPa).

Zhang et al. have described use of polyester sealing film—used for sealing of PCR well plates—for fabrication of multilayer microfluidic devices with advantages of low cost, biocompatibility, wide working temperature range, and optical transmittance in the visible region [130]. A CO_2_ laser was used for ablation of microchannels as well as alignment marks, and a custom alignment tool was used for alignment and bonding. Emaminejad et al. have demonstrated use of laser cutting and a tape substate for a wearable device that can be used for sweat collection, sample filtration, as well as biofluidic actuation and sensing [131].

### 3.3. Laser Fabrication

Lasers may scribe on a workpiece directly or through a mask and, consequently—in addition to fabrication of a mould for replication—can be used for machining of smaller volume microfluidic devices, which can include fabricating holes, channels, and complex 3D geometries, as well as joining and surface property modification [132,133]. UV and femtosecond (fs) lasers provide the best precision for machining of polymers, with resolutions for UV lasers to less than 50 μm in contrast to resolutions of several microns for fs lasers. The higher resolution for fs lasers does, however, come at the cost of higher investment and maintenance costs. The higher heat concentration on the substrate with CO_2_ lasers leads to a lower minimum resolution, which can be limiting for certain applications.

A microfluidic device for capturing circulating tumour cells (CTCs) was fabricated using a fs laser combined with micromilling and solvent-assisted assembly of the different layers [134]. Ni et al. [135] used a UV laser (AWAVE 355-15W-30K, Advanced Optowave Corporation, USA) with maximum laser power of 15 W and a laser beam wavelength of 355nm for cutting grooves within a polyvinyl chloride (PVC) film, which was then sealed by a commercial laminator to create a microfluidic device for inertial isolation of cancer cells from human blood. Mohammed et al. [136] used a CO_2_ laser engraving/cutting system (Trotec SP500, Australia) and a low power, multi-pass engraving with a solvent polymer reflow to reduce the imperfections and achieve to both small (50–500 μm) and large (> 500 μm) features within a microfluidic device. Li et al. [137] used a CO_2_ laser (ILS9.75, Universal Laser Systems, Inc., USA) to create capillary circuit components (trigger valves, retention valves, and retention bursting valves) within a PDMS substrate for sequential liquid delivery and sample reagent mixing. Gas-actuated microvalves and a peristaltic micropump [138] were created using an unfocused CO_2_ laser (VLS2.30, 25 W, wavelength 10.6 μm, Universal Laser Systems, USA) beam and surface treatment to create smooth semi-circular channels within PMMA and a flexible thermoplastic polyurethane (TPU) membrane that was thermally bonded between PMMA sheets. An IR (Infrared) laser was used to pattern graphitic carbon—with induced localised rapid pyrolysis—on polyimide (Kapton) film and create a flexible urea sensor through both direct urease enzyme immobilisation onto carbon and indirect electrodeposition of an intermediate chitosan before urease immobilisation [139].

### 3.4. 3D Printing

3D printing is a form of additive manufacturing technology where a three-dimensional object is created by successive layers of material. 3D printers are generally faster, more affordable, and easier to use than other additive manufacturing technologies. They offer product developers the ability to print parts and assemblies made of several materials with different mechanical and physical properties in a single build process. Advanced 3D printing technologies can allow for the development of product prototypes from the conceptual stages of engineering design through to early-stage functional testing and small volume production [140,141,142,143] in a quick and easy way. 3D printing also offers the potential to create microfluidic devices with fine features and at much lower cost than use of cleanroom type processes.

The worldwide sale of 3D printers has increased since 2003, and this has been accompanied by a decline in their cost. A number of competing 3D printing technologies are available; the main differences are in the way layers are built to create parts. Some methods use melting or softening material to produce the layers, e.g., selective laser sintering (SLS) and fused deposition modelling (FDM), while others lay liquid materials that are cured with various technologies. Fused deposition modelling uses a nozzle to deposit molten polymer onto a support structure, layer by layer. Another approach is selective fusing of print media in a granular bed. In this variation, the unfused media serves to support overhangs and thin walls in the part being produced, reducing the need for auxiliary temporary supports for the workpiece. Typically, a laser is used to sinter the media and form the solid. Examples of this are selective laser sintering and direct metal laser sintering (DMLS) using metals. Generally, the main considerations are speed, cost of the printed prototype, cost of the 3D printer, choice and cost of materials, and colour capabilities.

A basic method of 3D printing consists of an inkjet printing system, which are typically either based on a powder or photopolymer. The printer creates the model one layer at a time by spreading a layer of powder (plaster, or resins) and inkjet printing a binder in the cross-section of the part. There is minimal need for post printing finish work; the printer itself is used to blow off surrounding powder after the printing process. Bonded powder prints can be further strengthened by wax or thermoset polymer impregnation. Photopolymer systems use an array of inkjet printheads that deposit small volumes of build and support material in successive layers to form an object [144]. Resolution is given in layer thickness with X-Y resolution in dpi. Typical layer thickness is around 100 micrometres (0.1 mm), although some machines can print layers as thin as 16 µm. The X-Y resolution is comparable to that of laser printers. The particles (3D dots) are around 50 to 100 µm in diameter. We have previously described use of photopolymer inkjet 3D printing (i3DP) for creation of a microbioreactor with different functional layers joined by structured adhesive film [145] as well as fast and low cost prototyping of the mechanical and structural elements for a highly sensitive optical cavity enhanced absorption (CEA) detection system [146] (Figure 5).

Vat polymerisation—comprising stereolithography (SLA) and digital light processing (DLP)—are amongst the most widely used 3D printing approaches and use UV to cure resin and build a 3D printed object, essentially involving photochemical synthesis of materials. There are three key components: a light source as the energy input to activate some reactive species and allow reactions to occur; precursor materials with reactants, usually called photoinitiators, which are sensitive to light exposure; and a printing platform as a reaction container to hold the reaction mixture and maintain a certain reaction environment. SLA uses a UV laser beam that is directed towards a set of coordinates and hardens the photopolymer as it goes. The printing process therefore breaks the 3D design into multiple sets of coordinates for each layer. In the case of DLP, curing within a layer of resin is carried out by UV light from a DLP projector which remains stationary. The exposed liquid polymer hardens. A build plate moves in a small increment and the liquid polymer is again exposed to light. The process repeats until the model is complete. The liquid polymer is then drained from the vat, leaving the solid model. In DLP, the resin is cured one layer at a time, whilst in SLA, the laser beam moves from point-to-point tracing the geometry, and consequently SLA 3D printing is more accurate with a better quality of print. DLP is therefore preferred for printing parts quickly but with limited resolution, whilst SLA can provide higher resolution but is slower.

Ultra-small features may be produced by two-photon photopolymerisation. In this approach, the desired 3D object is traced out in a block of gel by a focused laser. The gel is cured to a solid only in the places where the laser was focused, due to the nonlinear nature of photoexcitation, and then the remaining gel is washed away. Feature sizes of fewer than 100 nm may be produced, as well as complex structures such as moving and interlocked parts. Microstereolithography (μSLA) is an extension of stereolithography and part of rapid prototyping. The system usually consists of a laser source, a tank of photocurable liquid polymer, a sample holder, and precision positional controls which move the relative locations of the laser and the polymer tank. The laser scans across the surface of the polymer tank according to the information of each cross section of the workpiece volume. Upon exposure to laser, the liquid polymer cross-links and solidifies. The sample holder moves the solid structure downward and allows for the construction of the next cross section; the whole structure can be built with successive layers. Complex structures can be built with this principle, and the minimum feature size is limited to the laser focus and the moving resolution of the positional controls. Certain metal and ceramic parts are also possible to be fabricated with this method with further curing and annealing processes.

The early μSLA systems utilised a point-by-point fashion to scan across each cross section. More advanced systems use projection instead of scanning of the laser to solidify the entire cross-section at once. Computer-controlled liquid-crystal display (LCD) and digital micro-mirrors (DMD) are used as projection means, or spatial light modulator (SLM), to direct laser beam; this function is similar to a photomask to create the features of each cross-section. The dynamically changed SLM has greatly increased the process efficiency. Other systems use two laser beams to create fine focus. The polymer will only solidify at the location where the two laser beams meet and provide enough energy intensity. The minimum resolution of these systems is in the hundred-nanometre range. An advantage with this setup is that the polymerisation takes place not only on the surface but also inside the polymer tank. Han et al. used a two-photon photolithography system (Nanoscribe 3D Photonics) to create master moulds curved structures, which is demanding, for complex liquid handling within a droplet microfluidic system [147]. Lu et al. have described fabrication of a micro-mechanical device using projection μSL [148]. The advantages of μSL lie in its short turn-around time; parts can be created in couple of hours, being convenient for manufacturing. A CAD file of the workpiece design can generally be used directly to control the movement between the laser source and the liquid polymer. No mould is necessary, and the process is simplified. The shortcoming of μSL is the limited available materials; only photocurable materials are feasible to be used for the process. Moreover, the surface is less smooth compared to other methods, and because μSLA is a serial process, it has been mostly used for prototyping or low-volume production and not typically for volume production [100].

3D printing remains at an early stage development and with considerable potential for new innovation for the development of novel microfluidic and sensing systems [149,150]. It has been used for fabrication of moulds that have been used for PDMS casting of microfluidic devices [151]. Folch et al. have demonstrated an interesting 3D printing approach for fabricating Quake-style microvalves and micropumps [152]. A key challenge for 3D printing has been to achieve microfluidic channels that are below 100 μm and with a footprint that is typically required for a variety of microfluidic applications. Although it has been possible to create channels that have width and height features that are below 100 μm, it has often been difficult to remove the resin/support material from the channel so that the length of channel is limited, which makes practical utility difficult. A very good characterisation of the practical microfluidic features that can be created for the Asiga Pico Plus with Pro3dure GR-10 has been described [153]. Separately for vat polymerisation, many of the resins can yellow over time, which can be an issue for transparency. Nordin et al. [154] have described a DLP-SLA printer with a custom resin, which has achieved impressive flow channel cross sections with a resolution of 18 × 20 μm. It has been shown that for DLP, better control over the resin polymerisation and quality of the microfluidic devices can be improved by reducing the penetration depth of the UV LED (Light Emitting Diode) light through a combination of wavelength selection and choice of absorber and photo-initiator materials and achieve 37 × 37 μm^2^ pixel resolution at a printed layer thickness of 25 μm with a modified desktop printer at 385 nm wavelength [155].

Gadegaard et al. [156] have demonstrated an interesting approach of 3D printing over a large volume with high resolution features by combining nanoimprinting with DLP—where the nanopatterns are created using electron beam lithography and reactive ion etching—and with the part being used as insert for injection moulding. Such an approach could allow higher volume manufacture of microfluidic devices which combine microfluidic channels with other elements such as gratings and fluidic filters. Hashimoto et al. [157] have described an interesting direct ink writing (DIW) 3D printer for dispensing a fast-curing flexible silicone resin on different substrates to create microchannels. Roppolo et al. [158] have reported 3D printing of photopolymer on the basis of PDMS towards fabrication of microfluidic devices, which can offer attractive optical, mechanical, and chemical stability features. There is increasing interest in development of 3D printing approaches that can allow use of multiple materials [159,160], which could be valuable for integrated valves, through a combination of rigid and flexible materials with different shore values, or integrated electrodes for electrochemical detection or fluid driving purposes. Lewis et al. [161] have demonstrated the potential for the integration of higher level of functionality through a novel hybrid 3D printing approach that combined direct ink writing of conductive and dielectric elastomeric materials with automated pick-and-place for surface mount passive and active electronic components for electronic circuitry.

## 4. High Volume Production

### 4.1. Hot Embossing

The hot embossing process typically transfers the mould features to a polymer substrate at an elevated temperature and pressure. The setup typically consists of moulds and inserts which contain the negative of features to be transferred, and the substrate polymer film to be processed. The process consists of three stages:The polymer film is inserted between moulds, and both the film and moulds are heated to or above the glass transition temperature of the polymer in a vacuum environment.The moulds are pressed against the softened polymer and the features are transferred.All parts are cooled to below the glass transition temperature of the polymer, and the processed polymer is demoulded (see Figure 6A).

Imprinting is a similar embossing procedure at room temperature. Elevated pressure is used to compensate for the lower mobility of the flowing polymer at lower temperature. This allows for the fabrication time to be shortened with the elimination of the thermal cycle. For hot embossing and imprinting, very fine features have been demonstrated in the range of several tens [162,163] to hundreds of nanometres [164,165,166]. When producing features in the nanometre range, this is often referred to as nanoimprinting.

Certain considerations are vital for the quality of hot embossed parts. First, a vacuum environment is essential, as any air trapped in the mould can create cavities in the embossed polymer or even ignite and burn the polymer under the embossing pressure. Becker et al. [11] points out that a vacuum environment may also help prevent corrosion of the nickel mould at elevated temperatures. A uniform temperature across the mould is necessary so that the internal stresses which cause warpage of the part produced may be minimised after demoulding. This is especially important when the produced part is large in size. The interaction between the mould and the polymer is important; low friction is necessary such that the moulded micro-structures and mould are not damaged during the demoulding process. A smooth and non-sticking surface of the mould and insert can reduce the friction during demoulding. A draft angle is beneficial to help reduce the friction of the demoulding process [11], especially for high-aspect ratio structures. Releasing agents can also help reduce the sticking between the polymer and the moulds if they do not interfere with later processes. Moulds for hot embossing are typically fabricated from either silicon or metal using microfabrication or CNC milling, respectively. Clearly, metal moulds are more durable and can be used for production of higher volume of parts, but it can be difficult to produce the smooth surfaces that would be necessary for joining of thermoplastic materials. Novak et al. [167] have described low-cost CNC milling of a positive mould into an acrylic sheet and casting in silicone to produce a negative mould which is mounted on a glass backing layer and used for hot embossing. Lillehoj et al. [168] have demonstrated that 3D printed metal moulds can be used for high quality replication of hot embossed microfluidic devices.

Several process parameters are important for the accuracy of a hot embossed part. The applied pressure, embossing temperature, and holding time of the moulds and the orientation of polymer chain are recognised. Higher applied pressure, higher embossing temperature, and longer holding time tend to reproduce micro-features closer to the mould [169]. Moreover, orientation of polymer chain in substrate is also considered as an effect on the fidelity of hot embossing. Jena et al. [170,171] have shown a variety of outcomes due to the directional alignment of embossed channels and material. Compared to micro-injection moulding, hot embossing has the following characteristics:Polymer has a lower thermal cycle since the glass transition temperature is lower than the melting temperature. The polymer also has shorter run into the mould because the material is on top of the mould directly instead of flowing through channels. Both aspects help reduce residual stress.The process uses lower pressure, lower flow rate, and a cooling rate of the polymer. As a consequence, hot embossed parts may achieve higher aspect ratio and smaller features. The finished parts tend to have smaller internal stress, which is particularly important for optical devices.The longer process time is due to the need to heat and cool both of the moulds and the polymer, as opposed to primarily the cooling cycle for injection moulding. The typical process times for hot embossing is between 4 and 15 min [11,100,172,173], but may be up to 30 min [169].

A shorter cycle time for hot embossing is highly desirable. The cycle time of the process is strongly affected by the tool design. Shorter process time can be achieved if improved heating or cooling channels are adapted in the mould insert and die, as is the case in injection moulding. A novel ultrasonic hot embossing has also been proposed to achieve localised heating of the polymer with a cycle time of few seconds [174,175,176]. Although only features in several tens of microns have currently been achieved, this process does offer a number of possibilities. Song et al. [177] used partial filling of micro-cavities on a mould insert, through surface tension and capillary action, to limit the surface imperfections on a microlens array. Ren et al. [178] have described an interesting approach of a one-step fabrication of polymer microfluidic devices where a Teflon negative mould is fixed on the upper heater of a hot embosser. The pre-heated upper heater with negative mould is lowered onto a PE membrane and a hybrid ethylene-vinyl acetate copolymer/polyethylene terephthalate (EVA/PET) membrane stacked coaxially on a PDMS-coated lower stage of the embosser set at room temperature. The membranes and channels are formed in the area that is not been pressed by the negative mould. The approach was able to fabricate microchannels with widths of 50 μm and obviated the need of separate step for sealing the channels.

### 4.2. Injection Moulding

Micro-injection moulding has the ability to produce large numbers of parts to small tolerance values with a high quality of surface finish. Hot melt polymeric material is injected at high pressure into a mould, allowed to cool, and then demoulded (Figure 6B). There are currently several injection moulding techniques available to transfer micron-scaled features from moulds to polymeric products [179,180,181]. Advantages of this production process include the replicability and cost effectiveness for mass scaling potential, especially for disposable products used for medical applications [182,183]; the availability of a wide range of thermoplastics; and it being fully automated, reducing cycle time [184]; as such, it is very attractive for research and industry for microfluidic applications [40]. The technique may be sub-classified into three broad categories:Micro-injection moulding: A modification of the standard injection moulding technique to produce small features or parts [13,185]. Micro-injection moulding seems to be the most suitable technique for microstructure replication with mass production scale potentials [179].Reaction injection moulding: This technique uses the combination of two polymeric components, one being a curing agent which is mixed and injected into a moulding tool [13,186].Injection compression moulding: A combination of injection moulding and compression moulding; it unifies the advantages of both manufacturing processes, and the de-moulding process is easier when compared to other techniques. Plasticised polymer is injected into a tool, and the mould halves are then pressed together to force the melt into the desired shape [13]. The moulded parts have high geometrical accuracy, narrow tolerances, high class surfaces, low residual stresses, and excellent mechanical properties. It has been found that the replication results improved when using injection compression moulding for features 10 μm wide and 5 μm deep [187].

There are several replication imperfections of the injection moulding technique. The nature of these replication imperfections differs according to the geometric levels of the mould cavity in use [188] and is classified at either macro or micro levels of imperfection.

Macro level of imperfection: Typical replication imperfections include shrinkage, warp, sink marks, flash, brittleness, burn marks, dimension variation, delaminating, black specks, etc.Micro level of imperfection: Replication at this level is dependent on surface topographic transcription between the mould and the part [188]. Prominent among the defects are slip, burst, and shrinkage, incomplete filing, etc.

The phenomenon of macro-level replication imperfection can be predicted analytically or numerically with certain level of accuracy, unlike those of micro-level replication imperfections, which are yet to be understood [188]. To achieve better replication of parts during the moulding process, one needs to optimise certain process parameters. Various studies show that among the prevailing process conditions the following are critical [40,179].

Mould pressure;Mould temperature;Injection speed;Injection pressure;Holding pressure;Holding time;Cooling time.

Optimising the process parameters requires careful examination of the overall process and making trade-offs, and the process is time consuming. However, it has proved to be useful in making some basic decisions and conclusions about the role of each of the parameters in aiding reproducibility of micro-features [179]. Different studies have attached different importance to process parameter combinations and have presented reasons for such importance. The main factors of injection moulding that have been investigated are melt and mould temperature, injection speed, and pressure due to their direct effects on the melt flow property. High melt and mould temperatures, and high injection speed have a positive effect on the melt flow in very small cavities [189].

Typically, microfluidic devices have large aspect ratio structures [190] and thin-walled geometries. The challenge is to completely fill the cavity before the solidifying layers begins to block the melt flow [179]. The part-to-part replication in mass production of small devices is also problematic. Accurate control of the process parameters such as the injection pressure, injection speed, mould temperature, holding time, and holding pressure [40] is needed to keep parts within specification. Elevating the mould temperature above the glass transition temperature of the polymer could enable a greater filling depth of the polymer into the features of the mould without the excessive use of injection pressure and speed with a trade off on cycle time, thus overcoming the hesitation effect caused by the viscoelastic nature of the polymer melt. Attia et al. [179] deduced that holding pressure provides better replication by offsetting part shrinkage at the expense of residual stress in the parts. Masato et al. [191] showed for replication of micro-pillars that a smaller cavity thickness of the main flow region, which allows quicker filling of micro-cavities before solidification, and a higher mould temperature results in higher replication. Cooling time and ejection pin configuration has also been shown to be a factor in part flatness. The interactive combination of processes parameters, material properties, and part geometry are yet to be substantially explored and investigated. The most appropriate means of investigating such interaction is statistical process control and the design of experiment (DoE) approach, which, due to the statistical nature of the method, requires a substantial number of runs in order to draw inference from the experimental findings [179].

Kim et al. [192] fabricated a micromixer with channel size of 250 by 60 µm from COC by injection moulding. The mould masters were produced by electroplating nickel through a SU-8 photo lithographically patterned mask. The pieces were thermally bonded to produce the mixer after the inlet holes had been drilled. Theoretical and experimental results agreed with one another on mixing efficiency, indicating that the reproducibility of the parts were sufficient. Mair et al. [193] produced a chip from COC with 100 by 100 µm channels and integrated interconnects for LIF (Laser Induced Fluorescence) measurements. The mould was electroplated in nickel and then polished to produce a finish suitable for the optical parts. The mould was held at 80 °C during injection of optimised parameters. Thermal bonding of the parts at 95 °C enabled the chip to withstand internal pressures of up to 15.6 MPa. Injection moulding has been used for fabrication of microneedle array from polycarbonate—using EDM for an aluminium master inlay inserted in a tool—which was subsequently functionalised to create a wearable device for minimally invasive continuous monitoring of key analytes [194].

### 4.3. Film or Sheet Operations

A recent trend in the fabrication of disposable microfluidic devices centres on using thin and flexible foil substrates. A thinner substrate is advantages for applications which require fast heat transfer, acceleration, or lower stiffness, because of the reduced mass and rigidity of the substrates. Typical examples include polymerase chain reaction (PCR), centrifugation, or flexible vibrating membranes. Foils can also provide protective and insulation functions for degradable contents and can be easily punctured to provide an easy mechanism for reagent transfer. From the cost perspective, the thin-film geometry requires less material; as Velten et al. [195] and Becker [196] have illustrated, the material cost per surface area dominates the total cost of a microfluidic device for large-volume manufacture. A foil-based approach is also convenient for borrowing of approaches from the longer established printing and packaging industries. The printing and packaging processes used for manufacturing are ideal for mass and scalable production to create affordable disposable devices. Several types of materials are available in foil form for the fabrication of microfluidic devices. The possible choices include polymers, metals, and papers. Additionally, the easiness to produce lamination of different materials offer further possibilities for foil-based systems. Liedert et.al [197] have described roll-to-roll manufacturing of disposable microfluidic devices. We focus the discussion here on polymer-based foil and use the definition of foils given by Focke et al. [198] of a flexible sheet thinner than 500 μm. The key aspects addressed here were mould fabrication and manufacturing using roller embossing and microthermoforming.

#### 4.3.1. Roller Embossing

Roller embossing is the patterning of thin films or foils by using circular rollers with moulds impressing the foil against counter rollers. The embossing rollers can be heated and become hot roller embossing. The foil is usually heated above the glass transition temperature before it is brought in contact with the rollers. Upon contact with the rollers, the foil takes the pattern, followed by cooling down and retaining the pattern upon leaving the rollers. Roller embossing or hot roller embossing can be viewed as variants of planar imprint or hot embossing for continuous manufacturing. Because rollers only contact the foil in the limited area where the two rollers are in contact, the smaller area makes it easier for both embossing and demoulding with lower applied force and heating up the material as well as a larger embossing area [99]. Moreover, hot roller embossing is operated in a continuous manner with high-throughput and can be combined with other operations such as printing for electrical contacts or lamination for encapsulation [93]. Makela et al. [199] demonstrated a continuous process with a gravure and a nanoimprint units on cellulose acetate film. Double-sided structure forming has also been demonstrated [92], although alignment remains an issue for further development.

A key challenge for roller embossing is to be able to transfer microstructure from a flat silicon mould to a large curved continuous roll. Striegel et al. [200] have described an approach for generating structure directly on a steel sleeve by coating the sleeve with a resist using a doctor blade, followed by UV-lithography on the curved surface, resist development, and electroplating of the structures directly on the sleeve to allow for creation of microstructure below 10 μm in thermoplastic polymers. Mead et al. [201] have demonstrated a flexible polyimide mould fabricated lithographically using a maskless Direct Write system followed by etching of features into a polyimide sheet.

It is difficult to achieve replicate accuracy of hot roller embossing, which is as good as planar hot embossing, since the material goes through a more rapid temperature ramp and a shorter contact time with the moulds. In hot roller embossing, the material exits the roller at about the same temperature as the roller and continuously changes its shape without a mould holding the material in place; the reflow effect can be more pronounced [94]. A shorter embossing contact time also means that the softened material may not have enough time to reach the deep recess of the mould. It has been shown that an aspect ratio of 1:1 is easy to replicate, 5:1 requires some care, and 10:1 can only be done with great difficulty [93]. The contact pressure at a localised area can induce stress and reduce the life of the mould. Moreover, the material from the initial embossing can be wasted before the optimal steady state of the machine is attained. To counter this problem, Velten et al. [97,202] devised a discontinuous system which is capable of stopping embossing before the optimal condition is reached.

The key parameters which affect the fidelity of the embossing, especially the features in the thickness direction of the foil, are the applied pressure, temperature, speed of the roller, and the pre-heating temperature of the foil. Higher pressure, temperature of the roller, pre-heating temperature, as well as a slower roller speed help to achieve deeper embossing depths [93,95,198]. Other effects such as the feature orientation with the rolling direction, pattern density, and hardness of the rollers are also being investigated [92,97,195]. Interesting effects such as non-symmetric profile and edges of the channels due to the direction of the roller should be considered carefully for particular applications [93]. Investigation on modifying the temperature profile of the embossed film with an aim to improve duplication fidelity by using a conveyer-belt mould is also worth noting [203].

There are variants of roller embossing methods. UV roller embossing uses UV-curable resin as the process material [89,90,91,204,205]. The material is firstly roller-embossed to acquire pattern and subsequently exposed to UV light for curing and setting the pattern. The technique does not require heating and cooling of the material, and lower pressure can be used; therefore, the process is ideal for device with temperature-sensitive reagents or proteins [198]. Pattern deformation is also less expected with this technique, with feature size down to couple of hundred nanometres [195,204,205]. The choice of the foil material can be limited because of the UV-curable property, such as foil thickness and optical property for the UV process and chemical compatibility for further processes [93]. Hesse et al. [206] have described a roll-to-roll (R2R) UV nanoimprint lithography approach with biofunctionalised channels for multiplexed DNA detection.

Another interesting variant is extrusion roller embossing [207,208]. This technique is a combination of foil or sheet extrusion and roller embossing. The hot extruded foils are directly fed into and pressed by the embossing rollers. Because the foils are already at a lower rigidity upon entering the contact of the rollers, heating of the material and rollers are not necessary, and the process can proceed at a higher speed. The shorter processing time is the major advantage of extrusion roller embossing. The extrusion characteristic of this technique can also be easily fabricated as laminated foils with other materials and possibly expand the application of roller process.

#### 4.3.2. Microthermoforming

Microthermoforming is a process that shapes thermoplastic films to acquire 3D structures. The film is clamped at its edges and heated to the glass transition temperature of the material; the softened material is then stretched against moulds with the assistance of pneumatic or mechanical pressure; the structure is then cooled and demoulded. Truckenmuller et al. [209] has given the variants on the microthermoforming process, which include micro matched-die moulding, rubber-assisted hot embossing, micro-backing moulding, and micro-pressure forming. In practice, the processes are very similar to that of hot embossing, other than forming with either compressed gas or vacuum. Both techniques use pre-processed polymer films or sheets, and the setups are largely interchangeable.

Despite the similarities of the microthermoforming and hot embossing techniques, there are, however, a number of differences [209]. Firstly, the film thickness in microthermoforming has to be smaller than the characteristic dimensions of the mould so that the film can wrap around the mould and become a conformal lining. In hot embossing, the film thickness is usually larger than the characteristic dimension of the mould because the material has to completely fill any cavities. Secondly, the film in microthermoforming is only softened and maintains its integrity. In hot embossing, the film is in a more fluid state so that the material can completely fill in the cavities. Thirdly, the film in microthermoforming is under tension because of the stretching, while the polymer in hot embossing is under compression due to the embossing effect. Finally, cavities in microthermoforming usually lead to hollow thin-walled structures, while cavities in hot embossing usually lead to solid structures. Variation in wall thickness is a characteristic of microthermoforming. The thicknesses are different along the base and top of a 3D structure, depending on forming into a positive (male) or negative (female) mould. This variation is the result of stretching and thinning-down of the film thickness around corners of the mould or along the drawing direction. The film on the side in contact with the mould usually has correct dimensions.

Microthermoforming is especially advantageous for 3D thin-walled structures, such as chambers and cavities, and work well as channels [210]. Formed thin sheet-based structures use less materials in contrast to the relatively bulky chip structures formed by hot embossing or injection moulding. Thin walls are manually deformable and are used frequently in blister packages for pills. The flexibility also makes demoulding easier than other moulding methods. The additional benefit of microthermoforming is the possibility of incorporating other processes to provide additional functions. A nano-textured structure on the sidewall of channels formed by microthermoforming has been demonstrated with a precedent nanoimprint process [211]; a porous 3D structure has also been demonstrated with first an ion radiation and later a chemical etch process [212]. Moreover, it is also relatively easy to seal the microthermoformed structures with another polymer film without damaging the microstructures.

A drawback of microthermoformed structures is that the characteristic feature size is limited by the thickness of polymer films because of the wrapping effect around moulds, i.e., a thick film cannot bend around conformally. The aspect ratio of mouldable structures is also limited by the film thickness, which can be stretched without breaking. The current practice seems to limit the ratio around one for negative forming [209], and three is possible for positive forming [198]. Microthermoforming can be a promising approach for mass production of microstructures with the optimisation of film heating and the pressure applying, as pointed out by Focke [198]. This is inferred from thermoforming in the macroscopic scale for commercial mass production of products such as disposable cups or containers. In these applications, the forming process only goes through a cooling cycle, instead of a heating and cooling cycle, as in hot embossing; the process time is therefore considerably shortened. Microthermoforming has been successfully used within centrifugal microfluidic systems with implementation of variety of unit operations and applications [213,214,215].

#### 4.3.3. Roll-to-Roll (R2R) Processing for Flexible Electronics

There is increasing interest within wearable and flexible electronics for diagnostic applications [216,217,218]. There are a wide range of examples, including skin-based sensors for wireless physiological monitoring for neonate and pediatric care [219], flexible sensors for wound management [220], DNA analysis using melting curve analysis with an integrated thin film heater [221], and impedimetric point-of-care diagnostic for selected biomarkers [222]. Integration of electronic functions can be based on printed organic semiconductor, or inorganic thin films or silicon chip devices which have advantages of functionality, performance, reliability, and power requirements. A key requirement is for patterning of metals for a variety of functions, including wiring to provide interconnections between the components, data transmission, electrochemical transduction, fluid driving, and heating, and for R2R processes, these can be conveniently carried out using screen printing or microfabrication processes.

Electronics manufacturing has largely been carried out using rigid printed circuit board (PCB) technology—having the advantages of reliability and low cost—and with increasing interest in use for the development of lab-on-chip systems [223]. An exception has been the development of radio frequency identification (RFID) tags, which are produced in very high volumes and at low costs and from the outset were developed using R2R processes. RFID tags represent an ideal use case for R2R processes since the coil antenna is larger in size than other integrated circuit (IC) chips—with an associated requirement for low cost substrate—and the IC itself only requires two electrical interconnects with little need for high precision bonding or high resolution metal wire interconnects.

Palavesam et al. [224] have described integration of multiple processes within R2R equipment, including sputtering or metal film on web substrate; web to web lamination e.g., lamination of dry film photoresist; lithographic patterning; electroplating of copper for fabricating wiring lines; screen printing of materials; chip bonding; and laser cutting. A temperature sensor label (Figure 7A) was demonstrated using a hybrid integration process which comprises screen printing, assembly of surface mount devices (SMD), and 3D integration of foils through lamination of pressure sensitive adhesive (PSA) tape for low-temperature gluing and use of laser to create via holes [224]. Figure 7B shows impedimetric point-of-care diagnostic cartridge for biomarker detection, which comprises an electrode layer using PEN substrate and a hot embossed fluidic layer and with both of these layers being joined using PSA tape that is laser-structured to open selected areas of the electrode layer [222].

Figure 8A shows semi-additive R2R processes [224] that can be implemented for patterning of metal on a substrate, and Figure 8B shows demonstration of producing the electrode layer for impedimetric detection described in [222] as a R2R process in copper, as a proof of concept. The process incorporates lamination of dry film photoresist, photolithography, development, etching, stripping of photoresist, and laser dicing and has the potential for extension to a wholly R2R manufactured microfluidic device.

## 5. Conclusions and Outlook

Polymeric microfluidic devices have attracted a large amount of attention. They can be fabricated as batch processes as well as volume manufacturing and have versatile material properties. The greatly reduced costs of polymeric films further enhance their roles in the manufacturing of disposable microfluidic devices and offers enormous potential.

Numerous fabrication approaches for polymers have been discussed, with each having characteristic features and a range of applications. For example, micro-injection moulding is ideal for volume manufacture but the material has to go through relatively wide process temperature; hot embossing can achieve very small details in the nanometre range, but their process time is relatively longer; thermoforming is ideal for making thin and 3D structures, but the minimum feature is restricted by the film thickness; e-beam or FIB undoubtedly can produce the smallest features among all methods, but their usual process size is probably around 100 by 100 mm, and the process time is relatively long for the production of one piece.

To benefit from the different available techniques, then, designers need to know the minimum feature sizes that these methods are able to achieve, along with a number of other parameters including the surface roughness, aspect ratio, and typical working size. These factors will lead to different fluidic outcome, such as pressure drop or mixing in the channels, as well as process time. There is therefore a need for scientists and engineers to have constant exchange of knowledge to achieve the optimised approach for a particular application. The situation can be further complicated when volume production and economic optimisation are considered for commercial purposes. As Becker pointed out, the total cost of a microfluidic device has to be broken down into design, fabrication, back-end process, and quality-control phases [225]. For a product that is produced in large volume, then approximately 80% of the cost comes from the latter two phases. Most academic research, however, is largely focused on the first two phases. Ideally the cost of each phase should be considered, whilst the product is still at the conceptual stage. Due to the availability of equipment and resources for individual projects, then it can be anticipated that each project may require further optimisation. There is a need to further elaborate general best-practice methodology as guidelines as well as greater standardisation for the fabrication of microfluidic devices [226].

Microfluidic devices have a wide range of potential applications including for point-of-care diagnostics [227], bioprocessing [228,229,230], and organ-on-chips for personalised medicine [231]. Molecular diagnostics requires considerable sample processing which needs to be integrated as different unit operations on the microfluidic device, including separation of blood or serum from whole blood before lysis, nucleic acid amplification, and amplicon detection. Combining individual functions and integrating multiple parts can promote the usability of these devices to be competitive to existing products. This approach will generally require integration of different manufacturing techniques, the understanding of their characteristics and limitations, and the proper arrangement of the manufacturing sequences. Such integration can realise a macro-machine with micro- or nanofunctioning parts fabricated in a continuous process, without additional machining or assembly. As an example, imagine a device with nano-, micro-, and macro-features to be possibly fabricated in several parts; hot embossed components for the nano-features followed by overmoulding to incorporate more complex functions with injection moulding for the micro-features; and eventually integrated and sealed by roller embossing and thermoforming in the macro scales. The scenario is very likely to happen after interface between individual processes are sorted out and standard fabrication sequences are established.

Of course, other non-polymeric manufacturing methods have to be incorporated into the process in order to achieve wider applications, such as deposition of electrodes or dispersion of reagents. These methods can enhance the functions of microfluidic devices and advance the application towards a complete and independent system. Furthermore, the trend to integrate the planar components into 3D structures is also a possible direction to increase the function and to reduce the size of microfluidic devices. Corresponding techniques, such as alignment, bonding, or surface modification, will need to advance with the development. Design of the device and its manufacturing processes can be more challenging for researchers. Overall, integration of all these techniques and the development of related processes seem to point out a promising direction for the wide usage of microfluidic devices. The possible capability of polymeric microfluidic devices will be almost unlimited.

## Figures and Tables

**Figure 1 micromachines-12-00319-f001:**
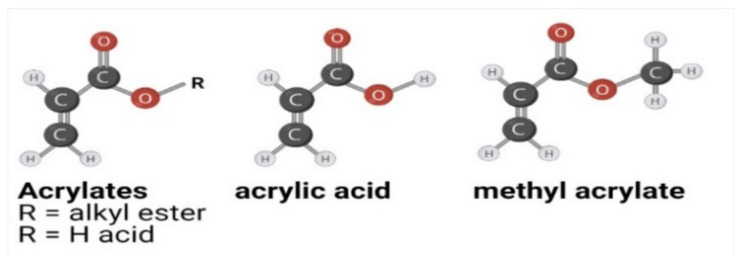
Structure for acrylates.

**Figure 2 micromachines-12-00319-f002:**
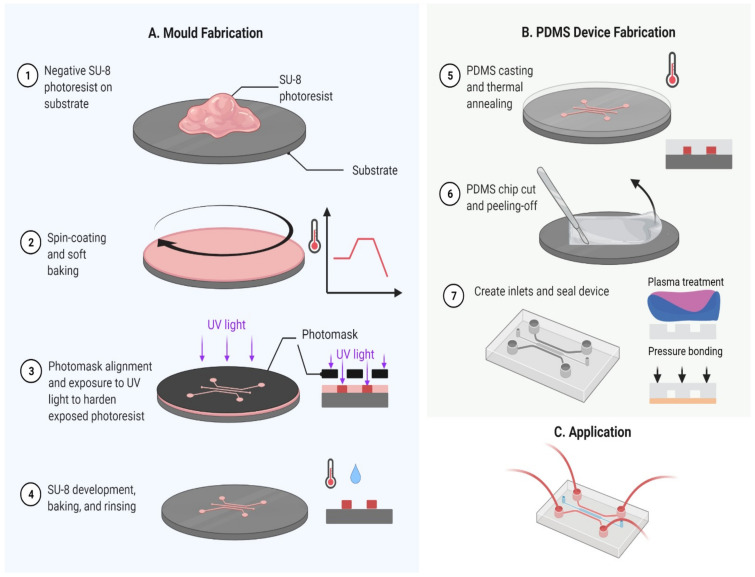
(**A**) Fabrication of an SU-8 master mould; (**B**) casting of polydimethylsiloxane (PDMS) on the mould, plasma treatment, and contact pressure bonding; (**C**) integrating PDMS microfluidic device with world to chip interfaces for application.

**Figure 3 micromachines-12-00319-f003:**
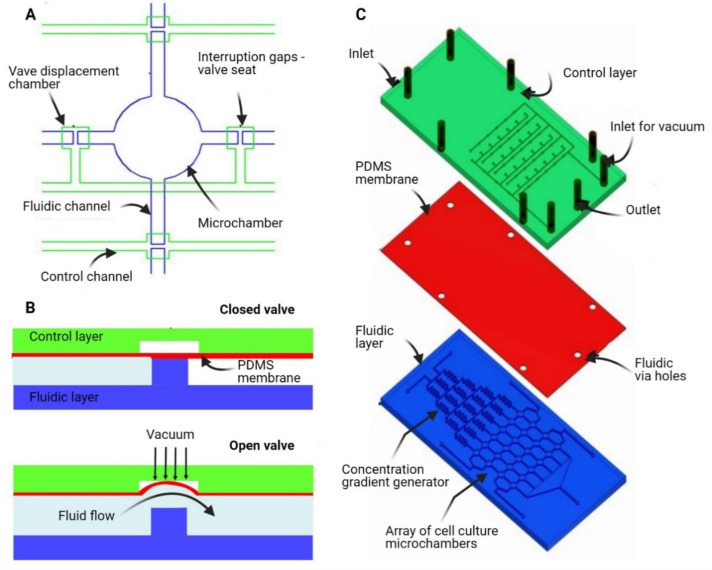
(**A**) Control of fluid flow in single cell culture element by two pairs of row and columnar valves; (**B**) opening and closing of valves by applying or releasing vacuum in the control channel to deflect or release flexible PDMS membrane; (**C**) 24 (6 × 4) element cell culture array with three-layer structure.

**Figure 4 micromachines-12-00319-f004:**
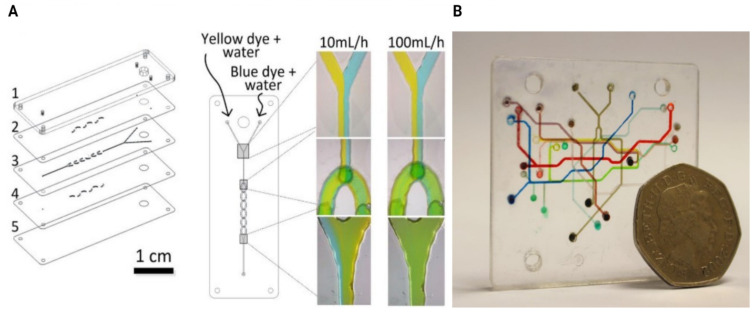
Laminated microfluidics created by stack of polymethylmethacrylate (PMMA) layers structured using laser cutting and joining using ethanol-assisted thermal bonding. (**A**) Five-layer split and recombine micromixer. (**B**) Nineteen-layer London underground map against a UK 50 pence coin from [129] under Creative Commons 4.0 International license.

**Figure 5 micromachines-12-00319-f005:**
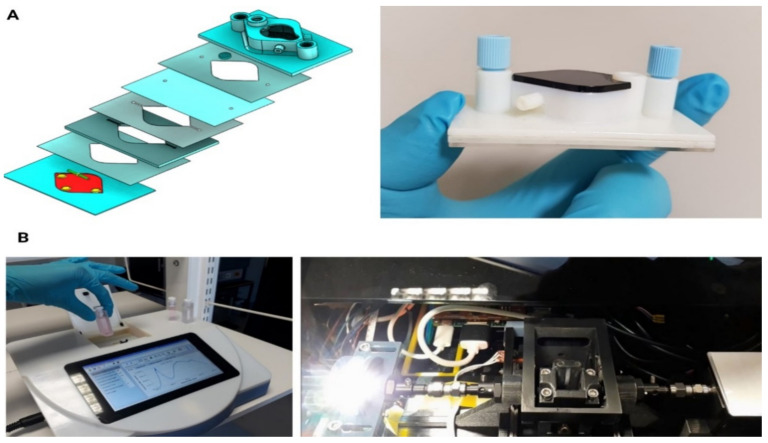
(**A**) Microbioreactor comprising i3DP fluidic and headspace layers joined by adhesive layers [145]. (**B**) i3DP mechanical and structural elements for cavity enhanced absorption (CEA) spectrometer [146].

**Figure 6 micromachines-12-00319-f006:**
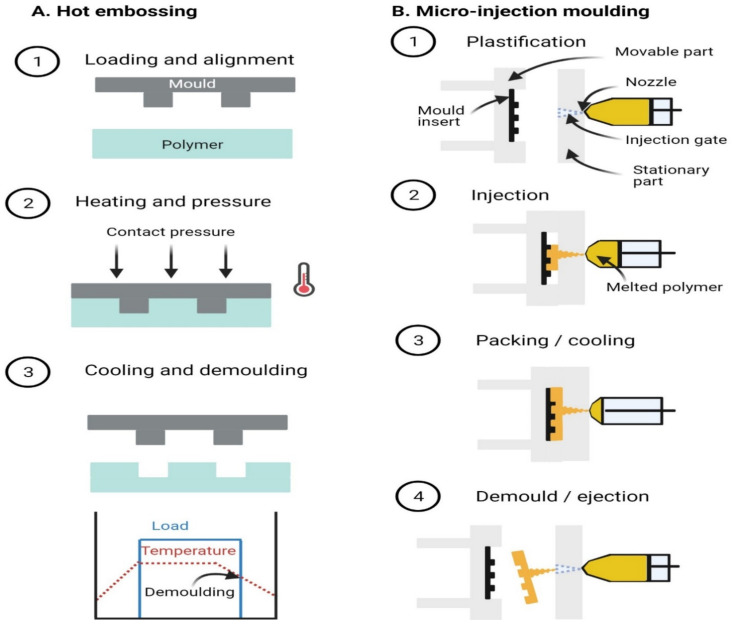
Schematic for (**A**) hot embossing and (**B**) micro-injection moulding.

**Figure 7 micromachines-12-00319-f007:**
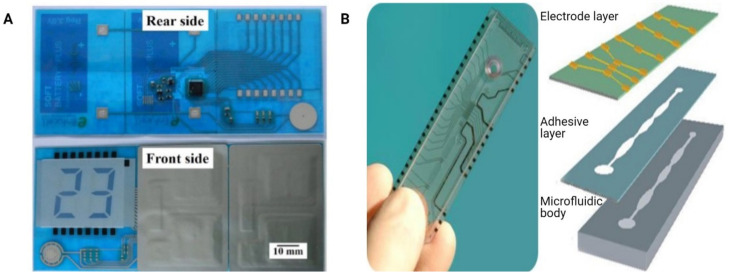
(**A**) Hybrid integrated temperature sensor label from [224] under creative commons 3.0. (**B**) Impedimetric point-of-care diagnostic cartridge described in [222] with electrode layer joined to microfluidic body using adhesive layer.

**Figure 8 micromachines-12-00319-f008:**
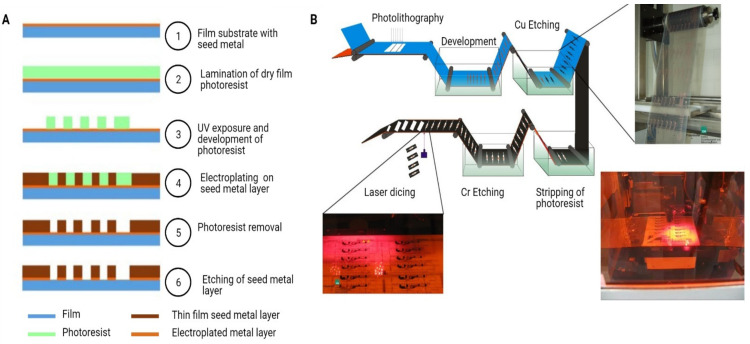
(**A**) Semi-additive roll-to-roll (R2R) processes for metal patterning from [224] under creative commons 3.0. (**B**) Schematic to show R2R processing, in copper, of impedimetric electrode layer in [222].

**Table 1 micromachines-12-00319-t001:** Comparison of master fabrication technologies.

Technology	Achievable Roughness Ra	Minimum Feature Size	Resolution/Feature Tolerance	Accuracy/Positional Tolerance	Typical Aspect Ratio	Typical Structural Dimension	References
Micro-cutting	65 nm	6.7 μm	2 μm	3 μm	<50	<1 mm	[15,16,20,26]
Ultrasonic machining	NA	5 μm	5 μm	NA	<7	NA	[15,16,20,21]
EDM	100 nm	5 μm	3 μm	1 μm	<20	<1 mm	[18,19,24]
ECM	28 nm	150 nm	5 μm	2 μm	<10	NA	[21]
Laser ablation	100 nm	<1 μm	≈1 μm	3 μm	<10	<500 μm	[15,20,21,27]
Focused ion beam	0.58 nm	40 nm	5 nm	100 nm	10	500 nm	[16,28,29]
E-beam	NA	10 nm	20 nm	NA	<2	<500 nm	[19,20,30]
X-ray LIGA	10 nm	50 nm	20 nm	300 nm	<100	<1 mm	[15,16,20,31]
MEMS process	10 nm	Some μm	NA	Some μm	<40	<1 mm	[20,32]
μ-SL	NA	<1 μm	120 nm	NA	NA	<1 mm	[21]

**Table 2 micromachines-12-00319-t002:** Excimer laser gases and wavelengths.

Laser Medium	F_2_	ArF	KrF	XeCl	XeF
Wavelength (nm)	157	193	248	308	351

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
