# Peer review of "Fabrication Methods for Microfluidic Devices: An Overview"

_micromachines, 2021, doi:10.3390/mi12030319_

Round 1
Reviewer 1 Report
This review paper is well-constructed and contains a large amount of information. However, part of the analysis is superficial, more analysis and newer articles is needed. I have some concerns to be addressed before I recommend this article for publication.
- Used the title "Fabrication methods for microfluidic devices: an overview" or something similar may be better. I think 99.99% of microfluidic devices currently in use are made of polymer.
- Most of the cited articles are very old, authors should read and cite more recent articles for a more advanced and in-deep investigation, for example, the3D printing technique can now print much higher resolution rather than what described here 0.1mm or 50um (Check these two links https://bmf3d.com/3d-printing-services/ https://2oec4j3twl711d5klt43pn63-wpengine.netdna-ssl.com/wp-content/uploads/2021/01/20-BMF-0014-Data-Sheet-printers-12521.pdf) https://doi.org/10.1016/j.eml.2019.100575
- The new fabrication technique such as two-photon polymerization should be included as a method of 3D printing and curve surface fabrication. it has the same resolution or even lower resolution than conventional photolithography. as a price of high-resolution the printing speed is low. So authors need to give a paragraph for this new fabrication method. here are some examples of two-photo polymerized devices. https://doi.org/10.1039/D0LC00757A;
https://doi.org/10.1007/s10544-020-00529-w
https://doi.org/10.1007/s10544-019-0466-x
Author Response
- Title has been changed as suggested by the reviewer but it should be noted that microfluidic devices for micro gas chromatography ( https://pubs-acs-org.ezproxy.tees.ac.uk/doi/10.1021/acs.analchem.8b01461) - which is an important category of devices particularly for environmental analysis - typically use combination of glass and silicon and have in no way been addressed by this review.
- Older papers have been included where these have been used as well as key seminal papers, further newer papers have been included. In respect of resolution of 3d printing, for practical utility one has to consider within the context of the footprint, difficulty of support removal, commonly available 3d printers and material constraints. The perspective expressed within the manuscript is not unusual, see Nordin et. al. (Annu. Rev. Anal. Chem., 2020, ref. 140) “…many manufacturers advertise <100-µm resolution, deliverable fluidic feature size is typically many times larger than that…”. A very good review of microfluidic features that can be created for the widely available Asiga Pico Plus by Beckwith et.al. (ref. 153) has now been included. We do not say that higher resolution cannot be achieved and have described novel work including that of Nordin et. al. for higher resolution devices with active components.
- A paragraph on 2-photon photopolymerization for ultra-small features exists in the section on 3d printing. Where the reviewer has requested a reference for inclusion and was accessible to the author then this has been incorporated into the manuscript.
Reviewer 2 Report
The authors present a review on the fabriction method of polymeric microfluidic devices. The topic is not very new, but It is treated in a new prespectives. The manuscipt is well written, but presents some major issues to be solved before publication.
1) At the end of the intro (before 1.1 section) I would add a brief summary of the paper, to help the readers to find the section of interest.
2) A lot of efforts are being done to improve the reconfigurability of the mould. The authors should be add a section on this in the sectiond evoted to the master mould fabrication. See e.g. 10.1016/j.jmapro.2018.07.030 etc etc
3) Section 2.2.3 treates the laser-based method for devices fabrication. The authors put this section in the part devoted to mould fabrication. However, the Laser ablation can be used both for rapid prorotyping and for fabricate the mold, as it is also stressed in the text "UV and femtosecond lasers provide the best precision for machining of polymers, with resolutions for UV lasers to less than 500m in contrast to resolutions of several microns for femtosecond lasers. ..." So, the auhtors should better adress the aim of this section. I suggest to split in two: the part about mould fabrication and direct ablation, putting the second one as a subsection of "3. Low-volume production", eventually adding other references about direct laser fabrication of microfluidic devices (10.1016/j.eng.2020.10.012, 10.1007/s10404-019-2206-1, ...).
Author Response
1) Additional comments have been made at the end of the introduction section (before 1.1 section) to help the reader navigate the paper.
2) Reference and comment has been included on the mould.
3) The laser based method section has been split into one section for mould fabrication and a second section for low volume prototyping. The suggested reference where they were available have been included.
Round 2
Reviewer 1 Report
Please read a few more times and correct the spelling and grammar mistakes.
Reviewer 2 Report
The paper has been improved and can be now accepted for publication in Micromachines.